# Systematic Analysis of Clemastine, a Candidate Apicomplexan Parasite-Selective Tubulin-Targeting Agent

**DOI:** 10.3390/ijms23010068

**Published:** 2021-12-22

**Authors:** Izra Abbaali, Danny A. Truong, Shania D. Day, Nancy Haro-Ramirez, Naomi S. Morrissette

**Affiliations:** Department of Molecular Biology and Biochemistry, University of California, Irvine, CA 92697, USA; iabbaali@uci.edu (I.A.); datruong@uci.edu (D.A.T.); sdday@uci.edu (S.D.D.); nharoram@uci.edu (N.H.-R.)

**Keywords:** plaque, ImageJ, Stress Fiber Network program, subpellicular microtubules, pyrimethamine, parabulin

## Abstract

Apicomplexan parasites, such as *Toxoplasma gondii*, *Plasmodium* spp., *Babesia* spp., and *Cryptosporidium* spp., cause significant morbidity and mortality. Existing treatments are problematic due to toxicity and the emergence of drug-resistant parasites. Because protozoan tubulin can be selectively disrupted by small molecules to inhibit parasite growth, we assembled an in vitro testing cascade to fully delineate effects of candidate tubulin-targeting drugs on *Toxoplasma gondii* and vertebrate host cells. Using this analysis, we evaluated clemastine, an antihistamine that has been previously shown to inhibit *Plasmodium* growth by competitively binding to the CCT/TRiC tubulin chaperone as a proof-of-concept. We concurrently analyzed astemizole, a distinct antihistamine that blocks heme detoxification in *Plasmodium*. Both drugs have EC_50_ values of ~2 µM and do not demonstrate cytotoxicity or vertebrate microtubule disruption at this concentration. Parasite subpellicular microtubules are shortened by treatment with either clemastine or astemizole but not after treatment with pyrimethamine, indicating that this effect is not a general response to antiparasitic drugs. Immunoblot quantification indicates that the total α-tubulin concentration of 0.02 pg/tachyzoite does not change with clemastine treatment. In conclusion, the testing cascade allows profiling of small-molecule effects on both parasite and vertebrate cell viability and microtubule integrity.

## 1. Introduction

*Toxoplasma gondii* is a eukaryotic, global human pathogen that causes birth defects, encephalitis, and blindness. *Toxoplasma* also serves as a model organism for studying conserved features of the microtubule-based cytoskeleton in other apicomplexans, which include the agents of malaria, babesiosis, and cryptosporidiosis [1]. Toxicity and teratogenicity of Daraprim, the standard-of-care treatment for toxoplasmosis, and the emergence and spread of artemisinin-resistant *Plasmodium* are two of many examples that illustrate the need for new safe and effective drugs that inhibit apicomplexan replication [2,3]. Microtubules are a well-established drug target for the treatment of cancer, gout, and helminth infection [4]. Tubulin and microtubules are an obvious but under-investigated target for antiparasitic agents to inhibit apicomplexan protozoan parasite growth.

CCT (chaperonin containing tailless complex polypeptide 1), also known as tailless complex polypeptide 1 ring complex (TRiC), is required for tubulin biogenesis, maintenance of microtubule stability, and restoration of GDP-dimer polymerization competency [5]. CCT folds nascent polypeptide of α- or β-tubulin into their native monomers. Tubulin cofactors then form tubulin heterodimers via the stable association between α-and β-tubulin proteins (Figure 1A). Heterodimer subunits reversibly assemble into microtubule polymers, which coordinate essential cellular processes, including mitosis, motility, and trafficking. Assembly-competent heterodimers bind GTP; subsequently, polymerization-dependent GTP hydrolysis in β-tubulin weakens lattice interactions to promote microtubule disassembly. The CCT/TRiC tubulin chaperone protein recycles GDP-bound tubulin dimers into the assembly-competent GTP-bound state. Because most microtubules are highly dynamic, pharmacological agents that perturb the balance between free tubulin dimers and polymerized microtubules block essential processes, including cell division, and cause targeted cell death.

For tubulin-targeting agents (TTAs) to be useful antiparasitic agents, they must be selective for the pathogen target with minimal non-specific activity or cytotoxicity to the human host. Anthelminthic benzimidazole drugs, such as albendazole, preferentially bind to and disrupt assembly of nematode but not vertebrate tubulins [6,7]. The potential of selective microtubule inhibitors as therapies for protozoan parasite infection is supported by evidence that dinitroaniline compounds disrupt microtubules in diverse protozoan parasites without affecting vertebrate host-cell microtubules [8,9,10,11,12]. Unfortunately, structure-function studies of dinitroanilines indicate that intractable properties of these molecules (hydrophobic and carcinogenic) make them poor drug leads [13,14]. However, their ability to distinguish vertebrate from protozoan tubulin demonstrates that identification and use of compounds that selectively affect protozoan parasite microtubules is practical. More recently, parabulin, a compound designed to distinguish the contours of the protozoan colchicine pocket over the vertebrate binding site, was shown to selectively target protozoan tubulin and inhibit the *Toxoplasma* lytic cycle [15].

*Toxoplasma* and other apicomplexans are obligate intracellular parasites that are vulnerable to TTAs during intracellular replication (Figure 1B). Compounds that selectively interfere with spindle microtubules block nuclear division in the dividing parasite, akin to the anti-mitotic effects of microtubules on cancer cells. In addition, apicomplexans use a corset of cortical microtubules to maintain cell shape; TTA treatment of parasites interferes with remodeling of these structures and the ability of altered progeny to invade into host cells [9,16]. Although individual apicomplexans grow in distinct cellular and organismal niches (with accompanying differences in metabolic pathways that often influence drug efficacy), the overall organization and pharmacological sensitivity of the microtubule cytoskeleton remains conserved in this group [1,17]. *Toxoplasma* is therefore well-suited to study tubulin, microtubules, and TTAs in a clinically significant and experimentally tractable apicomplexan parasite.

In this paper, we describe a testing cascade that was assembled to evaluate efficacy and selectivity of candidate parasite-selective TTAs in vitro. This methodology employs well-established techniques, such as plaque assays, cytotoxicity analysis, and detection of microtubules by immunofluorescent staining of vertebrate cells and extracted parasites. To transform qualitative assays into quantitative data, we leveraged open-access programs (ImageJ and SFEX) to capture statistics to describe microtubules in different treatment conditions [18,19]. These approaches permit us to corroborate qualitative data with quantitative analyses. In particular, we are able to discern shortening of *Toxoplasma* microtubules under conditions where this is not morphologically evident in intact parasites.

Here, we use this systematic analysis to evaluate the activity of clemastine (Figure 2A), an obsolete FDA-approved antihistamine that has recently been shown to inhibit replication of blood- and liver-stage *Plasmodium* parasites, agents of malaria and phylogenetic relatives of *Toxoplasma*. Prior work by others identified the *Plasmodium* tubulin chaperone complex TRiC-δ subunit as the primary clemastine target using thermal and chemical denaturation-based assays [20]. Pairwise comparisons indicate that *Toxoplasma* and *Plasmodium* TRiC-δ proteins are 63.5% identical and 80.3% similar (Appendix A) [21]. Selective TRiC inhibitors are of interest because they could synergize with direct microtubule-targeting agents to enhance inhibition of parasite replication. Targeting multiple proteins within the tubulin biogenesis and microtubule assembly pathway could prevent the emergence of drug-resistant parasites. In addition to clemastine, we also evaluated oryzalin, a well-defined selective inhibitor of protozoan tubulin, and astemizole, a distinct antihistamine with multiple mechanisms of anti-parasitic activity, including inhibition of heme detoxification in *Plasmodium* [22,23,24].

## 2. Results

### 2.1. Clemastine and Astemizole Selectively Inhibit Tachyzoite Growth

For antiparasitic therapies to be effective, they must selectively inhibit parasite replication with minimal vertebrate cell cytotoxicity. To assess this, we gauged the therapeutic index by measuring compound EC_50_ for *Toxoplasma* tachyzoites grown in vertebrate host cells versus the CC_50_ for uninfected human foreskin fibroblast (HFF) cells. Plaque assays were an established plate-based method to assess compound efficacy on *Toxoplasma* lytic growth [14,25,26,27,28]. Importantly, this method captures effects of drugs that act early in replication (i.e., dinitroanilines), as well as drugs that have a delayed or cumulative effect that requires several rounds of host-cell invasion and replication before becoming apparent (i.e., clindamycin). After 1 week of parasite growth, we visualized lytic destruction of crystal violet-stained host cells to reveal macroscopic plaques in the monolayer (Figure 2B). Greyscale images of individual wells were analyzed with the open-source program ImageJ [18] using the “threshold” functionality to differentiate between cells (dark gray) and plaques (white). The program calculates plaque number, average size, and percent total area within the field. The EC_50_ value is determined by graphing the relationship between compound concentrations and percent total plaque area relative to control cultures. Both clemastine and astemizole have an EC_50_ of about 2 µM (Figure 2C, Appendix A and Table 1). Consistent with previous data [14], the oryzalin EC_50_ for *Toxoplasma* RH tachyzoites is ~0.24 µM.

Parasite-specific tubulin-targeting compounds should impede *T. gondii* lytic-cycle stages that rely on proper tubulin and microtubule function, namely host-cell invasion and intracellular replication. To live image replicating parasites, we used a GFP-expressing *Toxoplasma* line [29] for invasion assays and replication assays. To confirm that GFP-expressing parasites have comparable sensitivity to wild-type parasites, we used plaque-based EC_50_ assays prior to use of this reporter line in other assays (Appendix A). The TRiC chaperone folds tubulin and actin [30], both of which are essential for tachyzoite invasion. To capture clemastine effects on protein folding that may reduce functional cytoskeletal components required for invasion, we assessed the effect of pre-treatment of intracellular (replicating) tachyzoites on subsequent infection efficiency. Treatment of RH-GFP parasites with oryzalin, clemastine (+/− pretreatment), or astemizole at their EC_50_ concentrations does not affect invasion (Appendix A). Next, we assessed the effects of the drugs on replication. By adding drug to infected cells after initiating parasite invasion, this assay measures replication independent of any effects on invasion. This methodology may also reveal details induced by drug treatment that are not resolved by the plaque assay, such as changes in parasite shape or acute replication delays. We enumerated the number of GFP-expressing parasites in individual vacuoles at 12-hour intervals over one complete lytic cycle. After tracking replication of the parasites treated with clemastine or astemizole for up to 36 h, we did not detect overt slowing of replication, such as was observed for parabulin [15], nor did we observe aberrant parasite morphology, which is seen soon after oryzalin treatment (Appendix A) [16].

### 2.2. Clemastine and Astemizole Vertebrate Cytotoxicity Is above Their Efficacy Range

To assess whether candidate drugs are toxic to vertebrate cells, we measured the 50% compound cytotoxicity (the CC_50_ value) using a commercial MTT assay (Figure 2D). Because replicating cells may be more vulnerable than confluent monolayers, we evaluated survival of both subconfluent (dividing) and confluent (contact-inhibited) HFF cultures after 72 h of exposure to increasing drug concentrations (Appendix A). Subconfluent cells do have somewhat lower CC_50_ values than confluent cells. Nevertheless, the tested drugs do not cause cytotoxic effects at or near the individual EC_50_ concentrations on either dividing or confluent cells. Using this information, we calculated the in vitro therapeutic index for these drugs, which is the ratio of the CC_50_ to the EC_50_. Oryzalin has the highest index, at 305, clemastine at 15, and astemizole has the narrowest index, at ~5 (Table 1).

**Table 1 ijms-23-00068-t001:** Test-compound efficacy and cytotoxic values.

Drug	*Toxoplasma* RH Strain	HFF CC_50_^subconfluent^	HFF CC_50_^confluent^	*Tg* Therapeutic Index Range	*Plasmodium* spp.
Oryzalin	0.24 μM	66.0 µM	78.3 µM	275–326	4.3 μM [31]
Clemastine	1.69 μM	16.3 µM	34.8 µM	9.65–20.6	1 µM [20]
Astemizole	2.07 μM	8.83 µM	18.0 µM	4.27–8.70	0.23 µM *Pf* RC [22] 0.086 µM *Pb* RC [23] 0.66 µM *Pb* EE [23]

Pf RC: *Plasmodium falciparum* red-cell stage. Pb RC: *Plasmodium berghei* red-cell stage. Pb EE: *Plasmodium berghei* exoerythrocytic (liver) stage.

### 2.3. Clemastine and Astemizole Shorten Toxoplasma Subpellicular Microtubules

To assess whether candidate drugs have selective anti-tubulin activity, we conducted a parallel evaluation of drug effects on both vertebrate and parasite microtubule networks. Vertebrate sensitivity is assessed using tubulin immunofluorescence to visualize microtubule networks after drug treatment of uninfected human HFF cells in culture (Figure 3A). To quantify microtubule polymer density between treatments, we used the SFEX program [19] to segment and enumerate filaments in a fixed-size region of interest. Treatment with the vertebrate microtubule-depolymerizing colchicine-site ligand combretastatin A4 [32] was used as a positive control to ensure that the SFEX parameters captured partial, as well as total, loss of cytoplasmic microtubules. Combretastatin A4 causes dose-dependent microtubule disassembly that the program accurately detects (Figure 3B). Samples treated with oryzalin, clemastine, or astemizole at or above the EC_50_ do not significantly differ from the vehicle control; vertebrate microtubules are intact and comparably dense.

In parallel, we assessed the effect of these candidate tubulin-targeting agents on parasite microtubules, also via tubulin immunofluorescence. Because tachyzoite subpellicular microtubules are unusually resistant to detergent extraction due to the association of SPM1 and other stabilizing luminal proteins, these structures can be specifically isolated for analysis [33,34]. Vehicle- and drug-treated parasites were incubated with deoxycholate, and the extracted subpellicular microtubule arrays were detected with tubulin immunofluorescence (Figure 4A). Images of parasite microtubules were measured using ImageJ. Subpellicular microtubules from vehicle-treated control cultures have an average length of ~4 μm (Figure 4B). Oryzalin-treated parasites exhibit a dose-dependent shortening of microtubule length, and we detected a significant reduction in microtubule length in tachyzoites grown in parabulin, a newly identified compound that selectively targets the protozoan tubulin colchicine site [15]. Tachyzoites grown in clemastine or astemizole exhibited smaller but significant decreases in filament length in comparison to untreated samples. To investigate whether microtubule shortening is a general consequence of treatment with drugs that induce parasite stress or slow growth, we quantified subpellicular microtubule length in tachyzoites treated with pyrimethamine, a well-characterized antiparasitic drug that inhibits folate metabolism by binding to the parasite dihydrofolate reductase [35]. Because microtubule length in pyrimethamine-treated samples is not different to that of the vehicle controls, we conclude that the shortening observed with either clemastine or astemizole treatment is not a general response due to stress induced by drug treatment.

This finding prompted us to investigate whether the reduced *Toxoplasma* subpellicular microtubule length observed in clemastine-treated samples is a consequence of decreased tubulin protein or reduced availability of polymerization-competent tubulin dimers in tachyzoites. To test this, α- tubulin protein levels in vehicle- and clemastine-treated *T. gondii* tachyzoites was quantified using immunoblots (Figure 4C). Parasite equivalents and known concentrations of porcine tubulin standards were resolved by SDS-PAGE for subsequent immunoblot analysis using a broadly cross-reacting α-tubulin monoclonal antibody that recognizes both vertebrate and *Toxoplasma* tubulin. Band intensity was quantified with ImageJ, and the porcine samples were used to create a standard curve to determine α-tubulin concentration in *Toxoplasma* samples normalized to parasite equivalents. Both control and clemastine-treated parasite samples contain ~0.02 pg of α-tubulin/tachyzoite, indicating that clemastine does not cause a decrease in total tubulin-protein concentration (Figure 4D).

## 3. Discussion

Tubulin and microtubules are essential for protozoan-parasite proliferation. As such, tubulin is an attractive drug target for the development of new antiparasitic therapies. In this paper, we describe an optimized testing cascade that delineates the activity of candidate TTAs on the apicomplexan parasite *Toxoplasma gondii* and on vertebrate host cells. We coupled assays that provide qualitative results with quantitative analysis programs to conduct unbiased candidate-TTA profiling. These assays evaluate drug effects on the parasite and host at both global (lytic growth) and target (tubulin) levels (Figure 5). We began by measuring the effective inhibitory concentration against parasites (EC_50_) and the cytotoxic concentration (CC_50_) against vertebrate cells with plaque and MTT-based cytotoxicity assays, respectively. To further investigate how compound activity influences aspects of the lytic cycle, we conducted invasion and replication assays to uncover phenotypes that cannot be discerned with plaque assays. Candidate TTAs are expected to impede parasite processes that rely on proper microtubule function. Importantly, cortical microtubules are non-dynamic in the absence of parasite-mediated disassembly, and extracellular parasites do not construct spindles, so we expect that TTAs will mostly influence intracellular replication rather than invasion of host cells. This is in-line with what we observe for oryzalin, which is a well-characterized protozoan microtubule inhibitor (Appendix A). It should be noted that once aberrantly shaped parasites arise in the presence of oryzalin, these forms are unable to reinvade into new host cells [16]. Following these analyses, we investigated the integrity of vertebrate and parasite microtubule networks post drug treatment. Vertebrate and parasite tubulin was visualized by immunofluorescence microscopy and analyzed using open access programs SFEX and ImageJ to quantify microtubules in control and treated conditions. The parasite tubulin analysis was supported by protein-level analysis using immunoblot data.

We and others previously described the selective disruption of protozoan-parasite microtubules by oryzalin and other dinitroanilines [8,12,13,14,36,37]. More recently, parabulin, a parasite-selective TTA, was shown to inhibit *Toxoplasma* tachyzoite replication [15]. As proof of concept for the testing cascade, we investigated whether clemastine indirectly targets *Toxoplasma* tubulin by impairing dimer folding and, in turn, limiting microtubule assembly. A prior study in the related apicomplexan *Plasmodium* demonstrated that clemastine binds to the TRiC chaperone that folds tubulin and actin, among other cytoskeletal proteins [20]. The authors of that study showed that clemastine treatment reduces *Plasmodium* tubulin levels and inhibits replication. Clemastine also inhibits the kinetoplastid parasites *Leishmania* (amastigotes), *Trypanosoma cruzi* (amastigotes), and *Trypanosoma brucei* (bloodstream forms) with EC_50_ values of 180 nM, 400 nM, and 3.7 µM, respectively [38,39,40]. The *Leishmania* study proposes inositol phosphoryl-ceramide synthase as the target: clemastine inhibits purified IPCS enzyme activity, with an IC_50_ of 2.90 μM; parasites with an IPCS gene knockout have reduced drug sensitivity, and clemastine-treated amastigotes show altered lipid profiles [38]. However, because the EC_50_ value for parasite replication is lower than the purified-enzyme IC_50_ value, clemastine may target more than one pathway. Given the *Plasmodium* data, we set out to assess whether clemastine disrupts tubulin-based processes in *Toxoplasma*.

In our analysis, we establish that clemastine inhibits the *T. gondii* lytic cycle, with a similar EC_50_ value to the *Plasmodium* data (Table 1). We did not detect a difference in the absolute tubulin concentration in control versus clemastine-treated samples as normalized to *Toxoplasma* parasite equivalents; both have ~0.02 pg of α-tubulin/tachyzoite (Figure 4D). This value encompasses all α-tubulin in the parasite: α-tubulin monomers, free dimers, and assembled microtubules. To corroborate our tubulin quantification, we calculated the expected amount of α-tubulin assembled in subpellicular microtubules in a single tachyzoite. The 22 subpellicular microtubules of untreated parasites are, on average, 4 µm long, giving a total length of 88 µm. Each subpellicular microtubule is composed of 13 protofilaments, and the rise of an individual tubulin dimer is 8 nm, yielding ~143,000 dimers. Each α-tubulin subunit has a molecular weight of 55 kDa, which allowed us to calculate that ~0.013 pg (~65%) of α-tubulin/tachyzoite is sequestered in subpellicular microtubules, with the remainder located in conoid, spindle, and intra-conoid microtubule structures, or as unassembled free dimers. Importantly, immunoblots do not distinguish total tubulin from the subset of polymerization-competent dimers in the cell. To evaluate this, we measured the length of subpellicular microtubules from immunofluorescent images of detergent-extracted tachyzoites. This assay detects a dose-dependent shortening of subpellicular microtubules from tachyzoites grown in oryzalin, a well-established direct microtubule-disrupting agent (Figure 3B). We recently characterized parasite growth inhibition induced by parabulin, a rationally designed parasite-selective colchicine-site ligand [15]. Although we could not discern a difference in the morphology of intact tachyzoites after growth in parabulin, we detected significant microtubule shortening relative to control samples. Pyrimethamine, a drug with a well-defined target (the parasite dihydrofolate reductase), does not reduce microtubule length, which indicates that shortening is not simply a non-specific consequence of exposure to any small molecule that induces parasite stress. Lastly, treatment with either clemastine or astemizole leads to a small but significant decrease in microtubule length.

The activity of astemizole was a surprise to us; we chose it as a likely negative control for the testing cascade. Astemizole is also an obsolete antihistamine that has been shown to target heme detoxification in *Plasmodium* parasites [22,23,24]. Because this pathway is specific to hemoglobin-degrading malaria parasites, we anticipated that astemizole would be inactive or less active against *Toxoplasma*. Although the *Toxoplasma* EC_50_ value is 10 to 20-fold higher than values measured for *Plasmodium* (Table 1), astemizole has a similar EC_50_ value to that of clemastine (~2 µM) and causes similar tachyzoite subpellicular microtubule shortening. More recent studies on astemizole indicate that it inhibits replication of liver-stage parasites, which are impervious to small molecules that interfere with heme detoxification, suggesting that there may be more than one mechanism of action. In a *Plasmodium berghei* model, the EC_50_ value for liver-stage parasites is 0.66 µM, approximately 10-fold the EC_50_ value for the red-cell stage (0.086 µM). A third antihistamine, hydroxyzine, has also been shown to inhibit *Toxoplasma*, with an EC_50_ value of 1 µM [41], but its target is yet to be uncovered.

The testing cascade described here profiles the efficacy and activity of candidate tubulin-targeting agents. We showed that the direct TTAs oryzalin and parabulin, as well as the indirect TTA clemastine, influence microtubule length. The degree of shortening detected after treatment with oryzalin or parabulin is larger than that induced by clemastine. The effects of clemastine are consistent with its inhibition of the TRiC-δ subunit of the tubulin chaperone. Surprisingly, astemizole also reduces microtubule length, albeit by an unknown mechanism, given that it does not interact with any defined tubulin targets. Given that astemizole reduces microtubule length to a similar degree as clemastine, it may be acting indirectly on tubulin polymerization, akin to the activity of clemastine.

We and others have identified small molecules that act directly or indirectly to influence tubulin and microtubule availability in parasites without off-target effects on host cells. Identification of favorable parasite-specific TTA leads may alleviate the limitations of available treatments, which are complicated by drug toxicity and the emergence of resistant parasites. We anticipate that the systematic analysis of candidate TTA compounds will be useful for other investigators to uncover leads for clinically relevant drugs. Ultimately, we hypothesize that using a combination of small-molecule inhibitors that directly and indirectly target protozoan tubulin should synergistically impede parasite growth. Moreover, *Toxoplasma* drug resistance is unlikely to occur if direct and indirect tubulin-targeting drugs are used together.

## 4. Materials and Methods

### 4.1. Parasite and Tissue Culture

HFF (human foreskin fibroblast) cells were grown in DMEM (Dulbecco’s Modified Eagle Medium) supplemented with 10% fetal bovine serum (FBS), 1% L-glutamine, and 1% penicillin-streptomycin [42]. Type 1 RH strain *T. gondii* tachyzoites and GFP-expressing [29] RH derivatives were serially passaged in confluent HFF cells in T25 flasks.

### 4.2. Drug Stocks and Use in Assays

Clemastine, astemizole, oryzalin, pyrimethamine, and combretastatin A4 stocks were obtained from Sigma-Aldrich (St. Louis, MO, USA). All compounds were dissolved in tissue-culture-grade DMSO (Sigma). In some cases, we compared pretreated and repeatedly (continuously) treated tachyzoites. In all, we compared the following four treatments: (1) untreated vehicle controls; (2) treatment alone (tachyzoites are exposed to drug during the assay); and (3) clemastine-combined pretreatment and mid-treatment (tachyzoites are exposed during the prior lytic cycle and during the experiment).

### 4.3. Determination of EC_50_ Values

HFF cells were grown to confluency in 6-well dishes and infected with 300 tachyzoites/well. Individual wells were treated with varying concentrations of drug, and the plates were left undisturbed for 7 days at 37 °C in a humidified 5% CO_2_ incubator. After one week, the plates were fixed (100% methanol, 5 min at room temperature), and a 5× crystal-violet histological stain solution was added to each well. Plaques appear as irregular clearings that interrupt the purple-colored host-cell monolayer. Greyscale photographic images of the plates were analyzed in ImageJ (NIH, Bethesda, MD, USA) using the “threshold” functionality. Plaque number, average size and percent total area were calculated by the program. To generate the EC_50_ curve, normalized percent total plaque area (relative to null controls) was plotted versus drug concentration using GraphPad Prism 9 (Graphpad Software, San Diego, CA, USA).

### 4.4. Invasion Assay

The red-green invasion assay using GFP-expressing tachyzoites was adapted from the original protocol [43]. HFF cells were grown to confluency on coverslips in 6-well dishes. RH-GFP parasites were harvested from infected host cells by syringe passage through a 27-gauge needle. Tachyzoites were allowed to adhere to host cells for 20 min at 37 °C in a high-potassium “Endo” buffer [44]. The buffer was replaced with warm invasion medium (DMEM + various drug conditions). Parasite invasion occurred under control or drug-treated conditions for 3 min (short invasion time) or 30 min (long invasion time) at 37 °C in a humidified 5% CO_2_ incubator. After invasion, samples were washed with PBS, fixed with formyl saline without subsequent permeabilization, stained with mouse anti-SAG1 antibody (Invitrogen, Waltham, MA, USA: D61S, 1:1000), and detected with a goat anti-mouse secondary antibody conjugated to Alexa Fluor 594 (Invitrogen A11005, 1:1000). Extracellular, invading, and intracellular parasites were scored from 10 random microscope fields. Invasion was calculated by dividing the number of intracellular parasites by the total number of parasites counted.

### 4.5. Tachyzoite Replication Assay

HFF cells were grown to confluency in 35 mm coverslip dishes (MatTek, Ashland MA, USA). RH-GFP *T. gondii* [29] were harvested by filtering freshly lysed parasites. Parasites were diluted 4-fold in D10+ media. After 30 min of invasion at 37 °C, the cell monolayer was washed with PBS to remove extracellular parasites. D10+ media alone, D10+ with DMSO, or drug in DMSO was added to the dishes. Actively growing parasites in fibroblasts were imaged every 12 h over a 48-hour period using a Zeiss Axiovert 200 M and an Axiovision camera to collect 10 random fields of view for each condition. Parasite replication states were manually quantified and tallied to enumerate singlets, doublets, quadruplets, and larger rosettes. Data are displayed as the fraction of total vacuoles/field that contain singlet, doublet, quadruplet, or larger (8+) numbers at the 36-hour time point.

### 4.6. MTT Toxicity Assay

For subconfluent monolayers, approximately 5 × 10^4^ HFF cells were plated into each well of a 24-well dish and exposed to D10+ media alone or D10+ with DMSO or drug for 3 days. The plates were processed with the In Vitro Toxicology Assay Kit, MTT based (Sigma Aldrich, TOX1), and A570 was measured in a SpectraMax I3X plate reader. Exported data were analyzed using Microsoft Excel. For confluent monolayers, approximately 5 × 10^4^ HFF cells were plated into each well of a 24-well dish and grown in D10+ media alone. Once the cells were confluent, they were gently rinsed with PBS, and D10+ with DMSO or drug was added to each well. The plates were processed as above.

### 4.7. Tubulin Immunofluorescence and Microtubule Quantification

HFF cells were seeded subconfluently onto sterile 12-mm circular glass coverslips in 6-well dishes. Cells were grown to semiconfluency (3–4 days). Cultures were exposed to DMSO (vehicle control) and drugs of interest for 60 min at 37 °C. These samples were fixed and permeabilized with ice-cold methanol for 3–5 min and blocked in 10% BSA in Ca^2+^, Mg^2+^-free PBS at room temperature for 30 min or overnight at 4 °C. Coverslips were stained with a mouse monoclonal anti-α-tubulin antibody (Sigma T9026, 1:2000) for 1 hour. After washing in PBS, coverslips were exposed to goat anti-mouse secondary antibody conjugated to Alexa Fluor 488 (Invitrogen A28175, 1:1000) for 1 hour. After washing in PBS, coverslips were briefly rinsed in dH_2_O and then mounted onto glass slides using VectaShield with DAPI. Images were collected on a Zeiss Axiovert 200 M. These images were exported and adjusted in Adobe Photoshop CC 2019. We initially explored using the SMLM image filament network extractor (SIFNE) program, which was written to analyze microtubule networks in images collected with super-resolution microscopy [45], but after consultation with its developers (National University of Singapore, Singapore), we used an alternative program (SFEX) that better accommodates wide-field tubulin immunofluorescence images. The open-source software Stress Fiber Extractor 1.0 [19] was used to quantify assembled tubulin (microtubules) present in HFF-cell immunofluorescence images within MATLAB R2015a, using the “Stress Fiber Network” analysis feature. For each treatment group, enhancement and segmentation was compared to the DMSO vehicle control. Quantitative analysis of the segmented filaments was performed with the Stress Fiber Extractor, with export of data into Excel (Microsoft, Redmond, WA, USA). Datasets were exported into GraphPad Prism 9 for statistical analysis.

To measure the length distribution of *Toxoplasma* tachyzoite subpellicular microtubules in control and treated parasites, we isolated subpellicular microtubule complexes by deoxycholate extraction of freshly lysed tachyzoites [33]. Briefly, RH parasites were treated with DMSO or candidate drugs in culture for 2 days. Following lysis of host cells, extracellular parasites were centrifuged at 1000× *g* for 20 min at 4 °C and then resuspended in 1 mL of PBS. Parasites were settled onto poly-l-lysine (Sigma)-coated coverslips for 15 min at room temperature. Coverslips were incubated with 10 mg/mL deoxycholate (DOC) in phosphate-buffered saline (PBS) for 10 min to extract the subpellicular microtubule cytoskeleton. Samples were fixed in 3.7% formyl saline (Sigma) at room temperature for 10–15 min, followed by immunostaining for tubulin, as described above. Extracted subpellicular microtubules in immunofluorescence images were measured using ImageJ.

### 4.8. Quantification of Tubulin Protein

To quantify tubulin levels in control versus clemastine-treated parasites, parasite lysates were compared to standards consisting of known amounts of purified porcine tubulin (Cytoskeleton, Inc., Denver, CO, USA). The samples were resolved on a 10% SDS-polyacrylamide gel (Bio-Rad) and blotted to nitrocellulose for immunoblot analysis. Blots were probed with a mouse anti-α-tubulin antibody (Sigma T9026; 1:5000) and detected with an IgG anti-mouse secondary antibody conjugated to HRP (Promega, Madison, WI, USA: W4021, 1:10,000). Samples were imaged using a Nikon D700 fitted with a 55 mm f/1.4 Nikkor auto-focus lens and intervening 12 mm Kenko auto-focus extension tube [46]. Exposure times were controlled with a Satechi TRC-B remote shutter control. Band intensity was quantified using ImageJ.

## Figures and Tables

**Figure 1 ijms-23-00068-f001:**
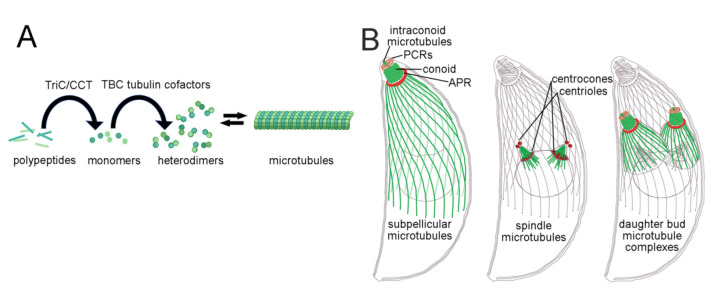
Tubulin, microtubules, and *Toxoplasma* microtubule-based structures. (**A**) Tubulin biogenesis requires TRiC-mediated polypeptide folding to form GTP-bound α- and β-tubulin monomers. TBC tubulin cofactors subsequently facilitate stable heterodimer formation. Genetic alterations or pharmacological inhibition of the TRiC/CCT chaperone reduces levels of functional tubulin subunits and influences microtubule dynamics. Tubulin-targeting agents shift the equilibrium between tubulin heterodimers and assembled microtubules to stabilize or destabilize microtubules. (**B**) To best illustrate the location and organization of the subpellicular, spindle, and daughter-bud subpellicular microtubules, each is shown individually, although all occur at the same time in replication of tachyzoites. *Toxoplasma* tachyzoites have an elongated shape imposed by association of a corset of non-dynamic subpellicular microtubules with the cytoplasmic face of the pellicle. These microtubules radiate out from the apical polar ring (APR), and a tubulin-based conoid can extend from or retract into the APR. The conoid is surmounted by two preconoidal rings (PCRs), and two microtubules traverse the conoid lumen. Intracellular parasites replicate using a closed mitosis coordinated by an intranuclear spindle to segregate chromosomes. Cytoplasmic spindle poles are associated with centrioles, and each hemispindle enters the nucleus through a specialized region of the nuclear envelope termed the centrocone. *Toxoplasma* replicates by endodyogeny, a form of internal budding. A daughter bud encloses each spindle pole. Buds consist of an inner membrane complex (IMC) associated with subpellicular microtubules and a conoid complex. Daughters emerge from the mother parasite by adhesion of the maternal plasma membrane to the daughter IMC that forms the mature pellicle.

**Figure 2 ijms-23-00068-f002:**
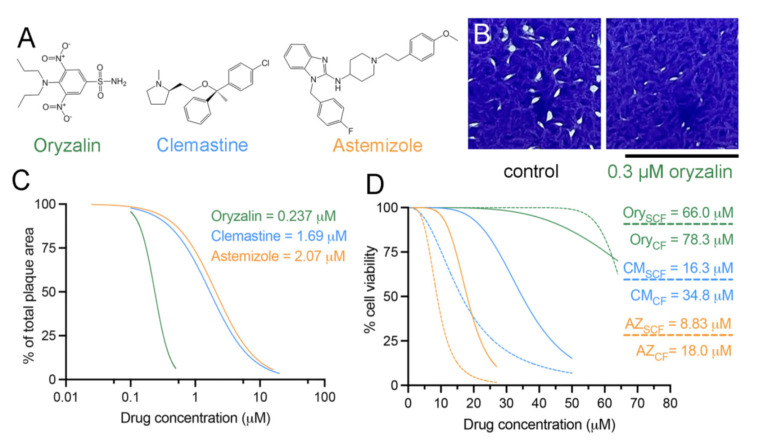
Efficacy and cytotoxicity establish the therapeutic index. (**A**) Chemical structures for oryzalin, clemastine, and astemizole. (**B**) After a week of undisturbed growth, lytic growth of RH strain tachyzoites creates visible holes in the HFF cell monolayer, as visualized after fixation, and crystal violet staining. Increased concentrations of compounds inhibit *Toxoplasma* replication and reduce the size and number of parasite-produced plaques. Control and 0.3 µM oryzalin-treated wells are shown. Scale bar = 1.75 cm. (**C**) The relationship between plaque area and drug concentration is graphed to determine the 50% effective concentration. All values are normalized to null control cultures. The EC_50_ values for clemastine (blue, CM), astemizole (orange, AZ), and oryzalin (green, Ory) showcase that all exhibit antiparasitic activity in the micromolar range. All results represent the average of at least 9 readings (3 biological replicates, each with 3 technical replicates) ± standard deviation. (**D**) The relationship between cytotoxicity (assessed by MTT assay-based quantification) and drug concentration is graphed to determine the CC_50_ values for confluent (solid lines, CF) and subconfluent (dashed lines, SCF) cultures of human fibroblasts. All results represent the average of at least 9 readings (3 biological replicates, each with 3 technical replicates) ± standard deviation.

**Figure 3 ijms-23-00068-f003:**
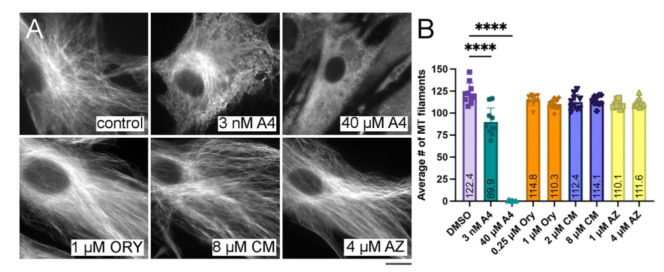
Quantification of vertebrate microtubules after drug treatment. (**A**) Tubulin immunofluorescence staining of HFF cell monolayers after 1 h of exposure to media containing the DMSO vehicle or drugs dissolved in vehicle permits visualization of tubulin subunits (diffuse staining) and assembled microtubules (filamentous structures). Scale bar = 10 µm. (**B**) Wide-field immunofluorescence images were analyzed with the open-source software SFEX to enumerate assembled microtubules by quantitative analysis of segmented filaments. The average number of microtubules in cytoplasmic regions of interest is only significantly different between the DMSO control and combretastatin A4 treatment, which partially or completely disrupts vertebrate microtubules. Statistical analyses were run using an ordinary two-way ANOVA. The results represent the average of 30 measurements (3 biological replicates with 10 fields of view for each treatment) ± standard deviation. Drug abbreviations: A4: combretastatin A4; Ory: Oryzalin; CM: clemastine; AZ: astemizole. ****: *p* values less than 0.0001.

**Figure 4 ijms-23-00068-f004:**
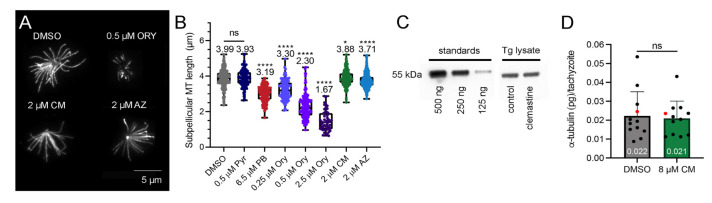
Quantification of *Toxoplasma* tubulin protein and microtubule length. (**A**) Tubulin immunofluorescence of drug-treated, DOC-extracted tachyzoites permits visualization of stable subpellicular microtubules. (**B**) Individual microtubule lengths were measured in ImageJ, and the distribution of lengths is shown for control and drug-treated samples. Statistical analyses were run using mixed-effects analysis. The results represent the average of 300 measurements (3 biological replicates with 100+ individual microtubules counted for each treatment) ± standard deviation. (**C**) Tubulin immunoblots probed with an anti-tubulin monoclonal antibody, detected with HRP conjugated secondary antibody. Protein samples contain defined numbers of *Toxoplasma* tachyzoites, and parasite tubulin was quantified relative to purified tubulin standards of known concentration. (**D**) Known porcine standard concentrations plotted against band intensity were used to calculate the concentration of α-tubulin in parasite samples. The red dots indicate the values for the blot data shown in C. Values were normalized to *Toxoplasma* equivalents, and statistical analyses were run using an unpaired t-test. The results represent the average of 12 measurements (3 biological replicates with triplicate or more technical replicates) ± standard deviation. Abbreviations: Pyr: pyrimethamine; PB: parabulin; Ory: Oryzalin; CM: clemastine; AZ: astemizole. ****: *p* values less than 0.0001; *: *p* values less than 0.05; ns.: >0.05, not significant.

**Figure 5 ijms-23-00068-f005:**
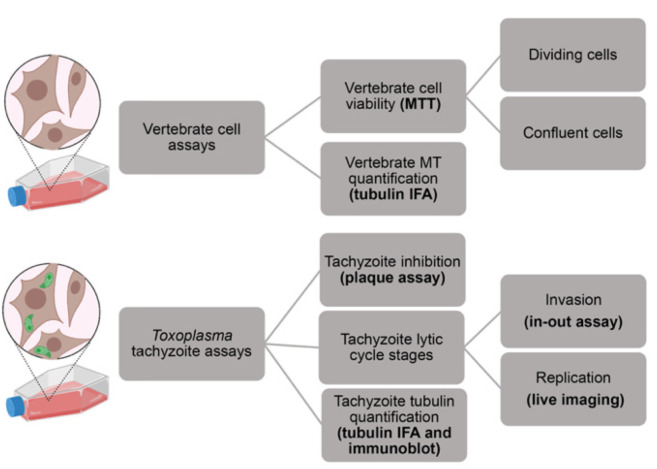
Parasite-selective microtubule-targeting agent testing cascade schematic. Analysis of compound effects on uninfected human fibroblasts capture any deleterious consequences to vertebrate microtubules or other key cellular processes. Assays using *Toxoplasma* tachyzoites cultured in human fibroblasts delineate drug efficacy and define effects on the lytic-cell stages of invasion and intracellular growth. Quantification of tubulin protein concentration and subpellicular microtubule length distribution identifies direct effects on tubulin and microtubules.

## Data Availability

Data is contained within the article or Appendix A.

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
