# Peer review of "Systematic Analysis of Clemastine, a Candidate Apicomplexan Parasite-Selective Tubulin-Targeting Agent"

_ijms, 2021, doi:10.3390/ijms23010068_

Round 1

Reviewer 1 Report

In this manuscript, authors developed an analysis pipeline to determine the efficacy and activity of candidate 343 tubulin-targeting agents. They claim that the direct TTAs oryzalin and parabulin as well as 344 the indirect TTA clemastine influence microtubule length. Clemastine’s effects are 345 consistent with inhibition of the TRiC-δ subunit of the tubulin chaperone. Surprisingly, 346 astemizole also reduces microtubule length, albeit by an unknown mechanism, given that 347 it does not interact with any defined tubulin targets.

The claims in this paper are consistent with literature and well supported by the data. I recommend this the publication of this manuscript in its present form.

Author Response

We thank reviewer 1 for their comments. We have checked grammar and spelling, and all are correct using American spelling. It is unclear how we could revise the introduction as it describes the parasite, its pathology, its use as a model organism to study apicomplexan microtubules and the prior work demonstrating that clemastine binds to the Plasmodium tubulin chaperone.

Reviewer 2 Report

The authors report the development of an in vitro assay to identify drugs targeting tubulin on Toxoplasma gondii and vertebrate host cells. Overall the report is good and focus on a import research issue for drug development for parasitic diseases. Nevertheless, the general perspective that the method can be applied for Apicoplexan parasite is very speculative since, as it stands, the application to other parasites rather than Toxoplasma are shown. I suggest authors to either show the application in other parasites or simply shaping the article describing it only to Toxoplasma and leaving the application to other parasites into the Discussion section.

Author Response

We thank reviewer 2 for their comments. We have checked grammar and spelling, and all are correct using American spelling. Toxoplasma gondii is a well-established model organism for studying the properties of cytoskeletal proteins of other apicomplexans which are more difficult to grow or image in the laboratory. See, for example the book “Toxoplasma Gondii: The Model Apicomplexan. Perspectives and Methods” (Kim and Weiss editors), Elsevier/Academic Press, 2019.

To demonstrate that Toxoplasma can be used to model the behavior of other apicomplexan tubulins, we include a Clustal omega alignment of beta tubulins from Toxoplasma, Plasmodium, Babesia and Cryptosporidium that illustrates the high degree of identity between them. The bulk of tubulin folds into a globular structure that harbors all known drug binding sites, so variability in the short carboxy terminal tail (after the YQQYQ sequence) should not affect drug binding. We include this alignment for the reviewer and if they would like to see it added to the supplemental data in the paper, we are happy to do so.

Reviewer 3 Report

I was not able to understand the purpose and aim of this article from the abstract, introduction and discussion.

I think the title does not match the content. If you want to make a statement about the analysis pipeline, you need to significantly alter the introduction and results text so that the reader can see it.

Also, the results and conclusions of this article are hardly worthy of publication in the IJMS.

Author Response

We thank reviewer 3 for their comments. It is unclear how we could revise the introduction as it describes the parasite, its pathology, its use as a model organism to study apicomplexan microtubules and the prior work demonstrating that clemastine binds to the Plasmodium tubulin chaperone. Without further information on how to improve the methods, we are not able to modify this section. We have deliberately included details on how we performed each aspect so to guide others who want to use a similar set of assays to assess candidate tubulin targeting drugs using Toxoplasma gondii.

We suspect that the reviewer’s concerns about our use of the term “analysis pipeline” may stem from the common use of this phrase to refer to computational analysis of complex bioinformatics datasets. Therefore, we have replaced the phrase “analysis pipeline” with other wording in the title and throughout the paper (see track changes).

Round 2

Reviewer 2 Report

The authors addressed the previous issues

Author Response

Thank you.

Reviewer 3 Report

The authors have replaced the phrase 'pipeline' throughout this article. I think this is a good improvement for many readers. However, there are still a few major aspects to be improved.

1. It is unclear what the point of this article is; systematic cascade or clemastine?

1.1. The authors should clearly state the conclusion of the results and significance of this article at the end of the abstract.

1.2. Although there is 'systematic cascade' in the title, the main points and flow of this are not described in main text. This could mislead the readers. If authors want to use this phrase in the title, authors should add them into the main text, especially in the introduction.

1.3. Also, although there is 'systematic cascade' in the title, this paper seems to be an evaluation of the effect of clemastine on Toxoplasma. Given the data presented in this paper, I think it would be more accurate to use a title that describes the effect of clemastine.

2. Some figure legends are not fully explained.

2.1. Fig. 2D; 'Ory', 'CM', 'AZ', 'SCF' and 'CF' should be explained.

2.2. Fig 3; 'A4', 'Ory', 'CM' and 'AZ' should be explained.

2.3. Fig 4B; 'Pyr', 'PB', 'Ory', 'CM' and 'AZ' should be explained.

3. Scale bar in Fig.

Fig 2B and 3A are missing scale bars.

4. References are not properly cited in the text.

References should be added on Line 226 (for parabulin), Line 336 (for parabulin), Line 352 (the more recent study on astemizole).

5. What is the meaning of '1-5-2' on Line 267?

Author Response

  1. The authors have replaced the phrase 'pipeline' throughout this article. I think this is a good improvement for many readers. However, there are still a few major aspects to be improved.

Thank you for this feedback. We address individual concerns below.

  1. It is unclear what the point of this article is; systematic cascade or clemastine?
    1. The authors should clearly state the conclusion of the results and significance of this article at the end of the abstract.

We understand that there is a disconnect between the title and the details of the paper. We hoped to detail the process of the systematic analysis using clemastine as a proof-of-concept compound. We have entirely revised the abstract to better convey this idea.

  1. Although there is 'systematic cascade' in the title, the main points and flow of this are not described in main text. This could mislead the readers. If authors want to use this phrase in the title, authors should add them into the main text, especially in the introduction.

We have added additional details on the systematic analysis to the paper text as well. See track changes.

  1. Also, although there is 'systematic cascade' in the title, this paper seems to be an evaluation of the effect of clemastine on Toxoplasma. Given the data presented in this paper, I think it would be more accurate to use a title that describes the effect of clemastine.

We have revised the manuscript title to better encompass both clemastine and the analysis. Our title is now “Systematic Analysis of Clemastine, A Candidate Apicomplexan Parasite-Selective Tubulin-Targeting Agent”.

  1. Some figure legends are not fully explained.
    1. 2D; 'Ory', 'CM', 'AZ', 'SCF' and 'CF' should be explained.
    2. Fig 3; 'A4', 'Ory', 'CM' and 'AZ' should be explained.
    3. Fig 4B; 'Pyr', 'PB', 'Ory', 'CM' and 'AZ' should be explained.

We apologize – the abbreviations were present in the abbreviation table and we have now added them to Figures 2-4.

  1. Scale bar in Fig.
    1. Fig 2B and 3A are missing scale bars

Figure 3A has a single scale bar under the panel which applies to all sub-panels. We have added a scale bar to Figure 2B and revised the legend.

  1. References are not properly cited in the text.
    1. References should be added on Line 226 (for parabulin), Line 336 (for parabulin), Line 352 (the more recent study on astemizole).

Thank you for noticing this omission. We have added the in-text citations.

  1. What is the meaning of '1-5-2' on Line 267?

We have deleted this detail because it is also in the methods section. It is the name of the specific anti-tubulin monoclonal antibody.

Round 3

Reviewer 3 Report

The authors have revised this article to make it easier to understand. The purpose and conclusion have also been clearly revised. I think that this is acceptable for IJMS.